# Effect of individual variations in genes related to dopamine brain transmission on performance with and without rewards during motor sequence and probabilistic learning tasks in children and young adults with and without cerebral palsy

**Barrett Dryden**[1¤], **Jesse Matsubara**[1], **Eric Wassermann**[2], **Hans Forssberg**[3], **Diane L. Damiano**[1] *

1 Neurorehabilitation and Biomechanics Research Section, Rehabilitation Medicine Department, Clinical Center, National Institutes of Health, Bethesda, MD, United States of America, 2 National Institute of Neurological Diseases and Stroke, National Institutes of Health, Bethesda, MD, United States of America, 3 Department of Women's and Children's Health, Karolinska Institutet, Stockholm, Sweden

¤ Current address: Department of Anesthesiology, Perioperative Medicine and Pain Management, Miller School of Medicine, University of Miami, Coral Gables, FL, United States of America
* damianod@cc.nih.gov

## Abstract

Children with cerebral palsy (CP) often participate in training to improve mobility, hand function and other motor abilities. However, responses to these interventions vary considerably across individuals even those with similar brain injuries, ages and functional levels. Dopamine is a neurotrasmitter known to affect motor skill acquistion in animals and humans and may be influenced by individual variations in genes related to brain transmission of dopamine. To evaluate potential genetic influences on learning in young people with and without CP, we calculated individual dopamine-related gene scores and compared these to the ability to learn two different tasks, an implicit sequence learning task and a probablistic classification task. Each task was also administered in an unrewarded condition and a rewarded one known to increase circulating levels of dopamine. The main finding was an interaction between gene score and condition for the sequence task such that those with low gene scores were poorer learners without rewards but responded positively to rewards whereas the converse was true for those with high gene scores. This is the first prospective study in CP suggesting that genetic variability may influence neurorehabilitation outcomes and could potentially be modulated using rewards or medications for those with poorer learning at baseline, thus promoting more personalized approaches to enhancing motor training in CP and other neurological conditions.

**Data Availability Statement:** Data will be available upon publication as doi:10.5061/dryad.qnk98sfs5.

**Funding:** This work was funded by the Intramural Research Program at the National Institutes of Health Clinical Center (Protocol # 16-CC-0149).

**Competing interests:** The authors have declared that no competing interests exist.

## Introduction

Cerebral palsy (CP) is the most common motor disability originating in childhood due to disturbances during early brain development [1]. Transforming the traditional view of CP as being caused by a specific perinatal injury, the etiology is often more complex and associated with a cascade of environmental risk factors [2]. More recently, multiple potential genetic factors have also emerged, which may also modify the risk for and severity of CP, as seen in other neurodevelopmental disorders such as autism [3–5].

Children with CP typically receive many different interventions during childhood to improve motor function. A growing body of evidence in CP provides guidance on which interventions are likely to be most effective to achieve specific treatment goals [6]. However, it is important to recognize that not all who receive an intervention will have the same response. Even motor training paradigms shown to be efficacious in more homogeneous groups of patients with CP, e.g. constraint-induced movement therapy in children with unilateral CP, show considerable variability in outcomes, with some showing large positive changes, others with incremental or no change, and still others with negative outcomes [7, 8]. The sources of this individual variability in treatment responses remain largely unknown, with conflicting findings across studies on whether factors such as younger or older age, greater or less neurological involvement, or certain brain lesions or compensatory reorganization patterns, are associated with better outcomes [8, 9].

Dopamine (DA) is a neurotransmitter with a major role in motor learning and plasticity as well in other aspects of brain functioning such as cognition and emotional/behavioral functioning. Tremendous progress has been made in basic neuroscience towards elucidating the complex neurobiological mechanisms by which DA modulates brain activity and the diverse biochemical processes that can be specifically targeted to alter its effects [10]. Variations in the genes that relate to dopamine function and neuroplasticity have been identified that have been found to affect motor learning in both healthy and clinical populations. It is possible that individual genetic variations may partly explain why children with CP, even those with similar brain injuries, show considerable variability in early recovery as well as the observed variability in responses to neurorehabilitation [7]. Disruptions in DA signaling are well-recognized in many neurological disorders, e.g. Parkinson disease [11], stroke [12], attention deficit-hyperactivity disorder [13] and traumatic brain injury [14], but have not been sufficiently studied in CP. Greater knowledge of potential genetic and/or disease-related differences in dopamine brain transmission could promote the development of more effective, individualized, neurorehabilitation approaches in CP and other brain disorders by identifying patients with a greater likelihood of positive responses to dopaminergic-related training modifications (e.g. rewards-based motor learning paradigms) or medications used to enhance training effects (e.g. L-dopa or methylphenidate).

Direct non-invasive measurement of DA transmission in the brain is difficult in humans but can be inferred by studying genes that regulate DA brain transmission. Pearson-Fuhrhop et al. [15] demonstrated that genetic variations in DA genes in healthy adults related to their ability to learn a novel motor task and further predicted differential responses to levodopa for enhancing learning. They randomized participants to a levodopa or placebo group during two weeks of motor training. No mean group difference in learning was found, suggesting that the drug had no effect. However, when participants were stratified according to a combined DA transmission gene score; those who had lower gene scores were found to be poorer learners on placebo and better learners on levodopa than those with higher gene scores who were conversely better on placebo and worse on levodopa. They concluded that an optimal level of DA is important for motor learning and the effect of the medication on an individual varies with

the baseline level of DA such that those with high levels may fare worse when given more (too much) DA. DA has complex interactions with other genes that modulate its transmission or affect behavioral outcomes and plasticity, e.g., catechol-O-methyltransferase (COMT) and brain-derived neurotrophic factor (BDNF, respectively). Their more common variants involving an amino acid alteration from valine to methionine in these genes (e.g., val158met; COMT: val66met; BDNF) also warrant investigation in CP [16–18]. BDNF has also been shown to have an important role in development, learning, and cortical plasticity [19, 20].

Our previous literature review [21] summarized data from clinical trials in non-progressive neurological disorders (stroke, CP, spinal cord injury, and traumatic brain injury) on the effectiveness of medications that altered DA transmission as well as data that related DA and plasticity-related genetic variations to motor recovery and responses to rehabilitation. Most medication trials were on adults post-stroke (16 of 24 studies) with positive results more likely in longer duration trials, but with the preponderance of data inconclusive. One small study [22] and one single case study [23] in CP reported positive motor effects from a short trial of levodopa. Two studies reported a direct association of higher gene scores with better motor recovery in stroke. One study [24] retrospectively recruited 33 participants with unilateral CP from earlier Constraint-Induced Movement Therapy trials to return for genetic analyses to determine individual gene scores as in Pearson-Fuhrhop et al. [15] and found that those with higher polygenic DA gene scores had greater post-training improvements on the Assisting Hand Assessment.

This is, to our knowledge, the first prospective study in CP evaluating the effect of DA brain transmission levels, indicated by gene scores, on learning abilities in children and young adults with and without CP. The first aim was to relate individual variations in genes related to brain DA and/or plasticity with the ability to learn two novel tasks in individuals with and without CP. The first task was a classic test of motor sequence learning, a serial reaction time task (SRTT) which involves both a motor (button press) and an implicit perceptual component (unconsciously learning a repetitive sequence of button presses). The SRTT has been used to evaluate learning among those with spinal cord injuries [25], Parkinson's disease [26, 27], basal ganglia lesions [28], and stroke [29–31]. The second was the Weather Prediction Task (WPT) which tests for increasing understanding of the probabilities of four cards in predicting one of two weather outcomes. The WPT has been used to evaluate probabilistic classification learning in adults with Parkinson's disease [32], Huntington disease [33], amnesia [34], and obsessive-compulsive disorder [35].

Our primary hypothesis was that variations in genes related to DA function and/or plasticity would be associated with variability in learning scores in individuals with and without CP. However, we anticipated that this association may not be as strong in CP given that DA levels may also be diminished due to hypoxia-ischemia [36] which is a common mechanism of brain injury in CP.

Rewards may enhance motor learning through an increase in DA brain transmission due to the pairing of reinforcement with the target behavior, [37] likely through the process of use-dependent plasticity [38]. A recent review on whether the use of rewards could enhance rehabilitation outcomes stated that data for the effectiveness of rewards were similar to the somewhat positive but inconsistent trends from DA-related medication trials [39]. The second aim of the study was to evaluate the effect of rewards on learning abilities and how this may be related to individual genetic variations. Each of the two above tasks were administered in an unrewarded and a rewarded condition. We hypothesized that rewards would have a mixed effect on learning for the group as a whole; however, those with lower gene scores would demonstrate poorer learning in the unrewarded condition and be more likely to show larger positive effects with rewards through the associated increase in DA brain transmission.

Conversely, we hypothesized that there would be a less positive or even negative response to rewards in those with higher gene scores. Also, given that those with CP may have lower DA transmission levels due to the brain injury, we anticipated that the group with CP may have a greater response to rewards.

## Materials and methods

### Participants

Inclusion criteria were diagnosis of CP or a healthy volunteer between ages 5 to 25 years, able to comply with the study protocol (e.g., follow instructions and have a blood draw for genetic analyses and attend two study visits). Additional criteria for participants with CP were being able to press keys on a computer with one hand, and not currently on levodopa, trihexypheni-dyl, methylphenidate, or baclofen as these may affect DA transmission or neuroplasticity. The protocol was approved by the Institutional Review Board at the National Institutes of Health (# 16-CC-0149). Written informed consent was obtained from participants 18 years or older and the parents of participants under age 18 with assent obtained from all minors. Participants were recruited from June 21, 2017 to September 22, 2021 with the large gap due to the COVID pandemic during which patient recruitment was suspended and resumption difficult.

### Procedures

Once enrolled, all participants underwent a blood draw for genetic testing. Based on results from exome-sequencing, individual genetic variants were determined for five genes implicated in DA brain transmission and learning and were used to calculate a gene score for each as described in Pearson-Fuhrhop et al. [15]. Brain-derived Neurotrophic Factor (BDNF) variations were also identified. Measures of performance in the two experimental tasks, the Serial Reaction Time Task (SRTT) and the Weather Prediction Test (WPT) were assessed, and both were administered during each of the two experimental sessions (rewarded and unrewarded), using a different version of each test across sessions.

### SRTT

The task and computer program used here were nearly identical to those used by Wilkinson et al. [40]. Participants were cued to push one of four computer keys corresponding to digits 2–5 of their dominant hand. The screen displayed four white boxes corresponding to each of the keys (see Fig 1). A figure of a dog appeared in the center of one of the boxes indicating which key to press. Participants were instructed to respond as quickly and accurately as possible. A trial ended when a key was pressed, at which time the figure disappeared. The next figure appeared after a short interval. The sequence of button presses was prescribed in advance with one sequence shown in 85% of trials (probable), and a second one interspersed 15% of the time (improbable). This task assumes that respondents unconsciously learn the probable

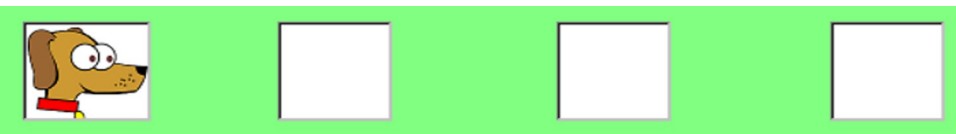

**Fig 1. Illustration of the figure projected on the computer monitor.** The figure of a dog appeared in one of the four boxes shown, corresponding to one of the four buttons on the keypad participants were instructed to press as accurately and quickly as possible once it appeared.

sequence but not the improbable one. There were 100 trials, analyzed in 20 blocks of 5 trials each.

During each session, either a rewarded or unrewarded version of the task was administered with the order randomized. In the reward-based feedback version, after each set of button presses, the participant received feedback in the form of a smiley or sad face associated with a small monetary reward depending on whether or not the number of correct responses improved during the current compared to the previous block. In the unrewarded version, they were randomly shown these same symbols, but these were not linked to performance or rewards. Participants were informed of this and told that they would still receive the standard amount of research reimbursement for their time, but no additional reward. For each version (rewarded/ unrewarded) of the test, data were reported for the probable and the improbable sequences yielding four different data sets or conditions: unrewarded improbable; unrewarded probable; rewarded improbable; and rewarded probable. Outcomes measures included the change in reaction time (from cue appearance to button press) and error rate (proportion of incorrect responses) over the 20 training blocks. After each test session, a questionnaire was given to test their Awareness of the probable sequence by asking them to choose the next number in the sequence from four possible choices.

## WPT

The Weather Prediction Task (WPT) is a well-established assessment of striatal or procedural learning capabilities and was originally intended to measure implicit probabilistic classification learning. It uses a set of four cards presented in different combinations from which participants are asked to estimate the probabilities of rain or sun or, alternatively, hot or cold. Each card has a different visual pattern with a fixed probability for a specific weather outcome. There are two strongly predictive cards with probabilities of 0.2 and 0.8 and two weakly predictive cards with probabilities of 0.4 and 0.6. On each trial, one to three cards with 14 possible arrangements were presented to the participant. The true probability for a specific weather outcome was determined by the cumulative probability of all the cards presented on a given trial and the two weather outcomes occurred at equal frequencies.

The WPT was administered here using two separate versions: a paired association (PA) unrewarded version and a feedback-based (FB) rewarded version using a specialized computer program developed by Dr. Leonora Wilkinson during her post-doctoral fellowship at our institution. In the PA version, participants learned probabilities through passive observation of different card combinations and their associated outcomes but were not asked to make predictions and no rewards were provided. In the FB version, participants predicted the weather outcome and received feedback in the form of smiley or sad faces that corresponded to a small monetary reward for a correct prediction. Each version consisted of 150 training trials followed by 42 testing trials in which participants made predictions during which no feedback or reward was given.

The two sessions (PA and FB) were completed at least one week apart in random order across participants. The association strength assigned to a specific card pattern differed across the FB and PA versions to prevent cross-learning across conditions. The FB and PA test scores reflected how well participants learned the associated card probabilities toward a particular weather outcome and were based on the proportion of trials in which they made a correct prediction for each condition. Test scores above 60% were considered significantly better than chance, indicating that learning had occurred. Reaction time, measured by how quickly they recorded their answer by pressing one of two keys, was also recorded.

At the end of each testing session, participants also completed a questionnaire to test their explicit Awareness of the card associations by asking them to predict what the card association strengths (probabilities) were from 0–100 for rain, 0 = definitely no rain to 100 = definitely rain. The Awareness score was computed by subtracting the true probability of the card from the value the participant gave and taking the absolute value, with a lower score representing greater knowledge of the cards' true probabilities.

## Genetic testing

DNA was obtained from blood samples collected from all participants by a trained phlebotomist or nurse into an EDTA tube. Samples were saved in a secured freezer at -80˚C for later batch analysis. Whole-exome sequencing was performed by Psomagen, Inc. (Rockville, MD). Bioinformatics analysis on the DNA-sequenced data was then performed and genetic variations were assessed for all participants with SNP analysis for COMT rs4680, DRD1 rs4532, DRD2/ ANKK1 rs1800497, DRD3 rs6280, DAT1 VNTR, and BDNF rs6265. These genes are identified as having variants that differentially affect brain DA transmission and/or cognitive learning.

A gene score was determined based on the genotype of each participant [15]. Polymorphisms in COMT, DRD1, DRD2, DRD3, and DAT1 that increase dopaminergic neurotransmission were given a value of +1, and those that decrease DA neurotransmission were given a value of 0. The individual genes scores were then added together for each participant to create a combined gene score between 0 and 5, in which 0 represents the least and 5 represents the greatest level of DA transmission. All genes were weighted equally. See S1 Table which describes the gene scores for each polymorphism. We also sub-grouped participants based on their DA gene scores into low (0,1,2) and high (3,4,5) to evaluate learning across low and high gene groups.

BDNF is a neurotrophin that is not directly associated with DA transmission, but instead with plasticity and neurotransmitter regulation. The BDNF polymorphism was analyzed separately and was defined as either 0 for Val/MET and MET/MET or 1 for Val/Val, with the assumption that the Val/Val polymorphism would be related to better learning.

## Statistical analyses

First, to evaluate whether learning of the sequence had occurred, a repeated measures general linear model (GLM) was used to compare changes in SRTT reaction time or error rate across the unrewarded probable and improbable conditions for all participants. We then repeated the above analyses first with the between-subjects group factor (HV or CP) and the two BDNF scores. We also computed change scores for each outcome and sequence type and then performed an independent t-test for change scores across groups.

The primary analysis to assess the influence of genetic variation on learning was a repeated measures GLM to compare learning on the SRTT by high and low gene groups (between subjects) and across unrewarded and rewarded probable conditions (within subjects). The task Awareness score was tabulated and correlated with the SRTT RT and Error Rate results to determine whether explicit knowledge of the sequence was related to their performance. All analyses were repeated for the WPT data.

We correlated individual DA gene scores to SRTT and WPT outcomes using Pearson *r* correlation procedures for all participants, as well as for the control and CP groups separately. To evaluate whether there was a speed-accuracy tradeoff in SRTT outcomes, Pearson correlation procedures were used to relate the change in reaction time to the change in error rate for the unrewarded, probable sequence trials (this condition was the baseline measure of learning the

sequence). We also related performance across the SRTT and WPT tasks to evaluate links between these two different types of tests, each of which was deemed to be related to DA brain transmission.

Secondary analyses were performed to evaluate the potential confounding effects of condition order, age and sex as a biological variable on outcomes using independent t-tests. We also correlated age with all unrewarded outcomes using Pearson $r$ procedures.

## Results

### Participants

A total of 29 participants were enrolled, 15 with CP and 14 healthy volunteers (HV). All underwent a blood draw for genetic analysis. Two participants withdrew due to inability to complete the tasks in the first session: one in the HV group and one with CP who had significant upper limb motor involvement as well as mild cognitive impairment. Two participants, one with CP and one HV, were excluded due to inadequate quality of their genetic data. In total, 13 participants in the CP group (14.6 years ± 5.1) and 12 in the HV group (15.2 ±4.7) with no significant age difference between groups (p = 0.80) completed the study and were included in the analysis. Participants with CP were also classified by hand function using the Manual Ability Classification System [41] and mobility function using the Gross Motor Function Classification System [42], with Level I indicating the highest function and Level V the lowest function. See Table 1 for participant descriptions.

### Serial reaction time task results

**All participants: Unrewarded probable vs. improbable sequences comparisons.** Fig 2A and 2B show the change in reaction time and error rate, respectively, for the two SRTT conditions (unrewarded, rewarded), and the two sequence types within each condition (probable, improbable). Change scores for probable trials are also presented in Table 2. Learning on the SRTT was indicated by faster reaction time and/or lower, or less of an increase, in error rate, on a probable compared to an improbable sequence. In the unrewarded condition, the mean reduction in reaction time was 174.2 ms for the probable sequence and 83.7 ms for the improbable sequence. A significant main effect was observed for sequence type (p = 0.02) signifying that improvement had occurred with training and was greater for the probable sequence, as expected. There was no significant interaction. The mean change in error rate was +8.2% for the probable sequence and +2.4% for the improbable sequence. A significant main effect was observed for sequence type (p < .001) and time (p < .001) as the error rate *increased* over time, and to a greater degree in the probable condition, both opposite to what we had anticipated. However, since reaction times improved with training and errors increased, we decided post hoc to correlate these two values to determine whether there was a speed vs accuracy trade-off. For the unrewarded probable trials, we found a significant correlation showing that greater improvement in reaction time was associated with a greater increase in error rate (r = 0.59; p = 0.008), indicating that these were inversely related and suggesting that participants may have prioritized improvements in reaction speed at the expense of accuracy in the SRTT.

Participants were further tested for Awareness of the sequence by guessing the next number in the sequence by identifying the next location of the dog's head (1 of 4 possible choices) when presented with part of the sequence. Mean Awareness Score was 4.6 ± 2.0 (12 is highest score), which although it was almost twice more than chance, it was not correlated with RT or Error Rate performance.

**Table 1. Characteristics of participants.**

| Subject ID | Age | Sex | Dominant Side | CP Sub-type | GMFCS[a] | MACS[b] |
|---|---|---|---|---|---|---|
| CP1 | 16.0 | F | L | Bilateral | I | I |
| CP2 | 14.3 | F | L | R. unilateral | I | I |
| CP3 | 14.3 | M | L | Bilateral | II | I |
| CP4 | 17.3 | F | R | Bilateral | III | I |
| CP5 | 6.6 | M | R | L. unilateral | II | II |
| CP6 | 14.9 | M | R | Bilateral | II | III |
| CP7 | 14.5 | F | R | Bilateral | II | III |
| CP8 | 14.0 | M | L | Bilateral | II | I |
| CP9 | 22.0 | F | L | Bilateral | II | I |
| CP10 | 25.7 | F | R | Bilateral | II | I |
| CP11 | 10.2 | M | R | L. unilateral | I | I |
| CP12 | 8.3 | F | R | Bilateral | I | I |
| CP13 | 12.1 | M | R | Bilateral | III | I |
| HV1 | 12.8 | F | L | | | |
| HV2 | 17.3 | F | R | | | |
| HV3 | 14.9 | M | R | | | |
| HV4 | 15.9 | F | R | | | |
| HV5 | 15.2 | M | R | | | |
| HV6 | 22.5 | F | R | | | |
| HV7 | 13.7 | M | R | | | |
| HV8 | 25.1 | F | R | | | |
| HV9 | 13.8 | F | R | | | |
| HV10 | 8.1 | F | R | | | |
| HV11 | 10.3 | M | R | | | |
| HV12 | 12.5 | M | L | | | |

[a] GMFCS = Gross Motor Function Classification System

[b] MACS–Manual Ability Classification System

**CP vs. HV: Unrewarded probable vs. improbable sequences comparisons.** Table 2 presents the changes in reaction time and error rate for the group as a whole, then separately for the two participant groups with the associated p-value for the independent t-test. A notable but nonsignificant trend was observed for group (p = 0.058) suggesting that those with CP tended to learn the probable sequence to a greater degree than controls. There was no significant interaction between sequence type and group (p = 0.60). For error rate, a significant main effect for group was observed (p = 0.03) with the control group demonstrating fewer errors. A significant sequence type by group interaction was also observed (p = 0.03), with the control group showing similar error rates across sequences, whereas the group with CP had a relatively higher error rate in the probable sequence.

**All participants: Unrewarded vs. rewarded probable sequences comparisons.** The second aim was to evaluate the effect of rewards on performance. Even though the mean improvement in reaction time was greater in the rewarded compared to unrewarded probable conditions, reaction time and error rate changes did not differ significantly between conditions for the group as a whole (p = 0.37 and 0.84, respectively).

**CP vs. HV: Unrewarded vs. rewarded probable sequences comparisons.** When adding the group factor, a significant between group difference was found for error rate change (p = 0.03) but not for reaction time change (p = 0.21). Those with CP showed markedly greater

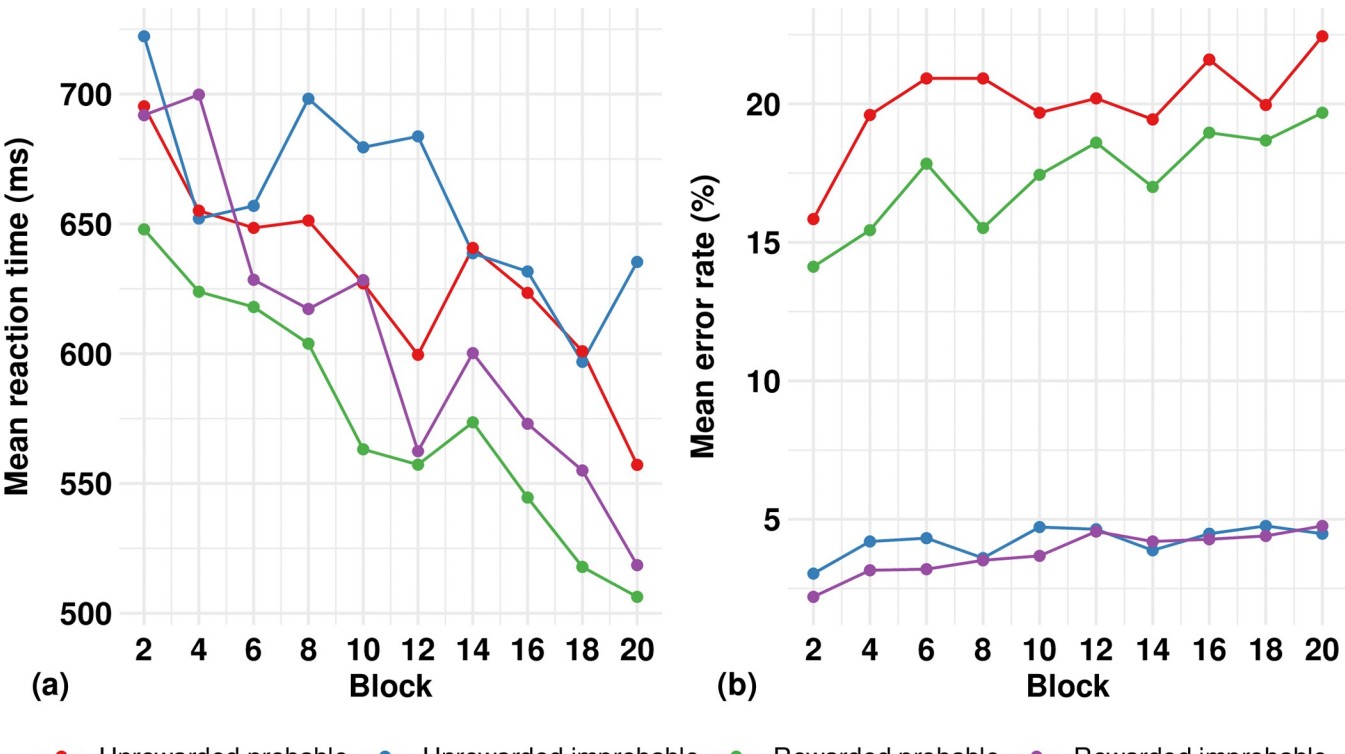

**Fig 2.** a. **Mean change in reaction time across training blocks for the serial reaction time task (SRTT).** Rewarded trials had generally faster reaction times post training than unrewarded ones, and that the probable sequences within conditions had faster reaction times within conditions post training compared to improbable sequences. b. **Mean change in error rate across training blocks for the serial reaction time task (SRTT).** Error rate increased more over time in the unrewarded trials, particularly for the probable sequences. Error rate was lower with minimal increases over time in the rewarded trials.

error rates than the HV group. There was no significant difference between the rewarded and unrewarded conditions (reaction time: p = 0.39; error rate: p = 0.85) or a group by condition interaction (reaction time: p = 0.73; error rate: p = 0.85). See S1A and S1B Fig. which show data for these conditions across groups.

## Weather prediction task

Table 2 also presents the WPT scores for the whole group and subgroups. Mean scores for the entire sample for proportion correct were 73.5 ± 13.9 for the Paired Association (PA; unrewarded) condition and 67.3 ± 17.4 for the Feedback (FB; rewarded) condition, with both greater than 60.0% which was the threshold for being greater than chance. Independent t-tests showed no significant differences between the CP and HV groups on the FB (p = 0.25) or the PA proportion correct score (p = 0.62). Interestingly, the mean score was higher for both groups in the PA test compared to FB but even with data from both groups combined, the difference between the two conditions did not reach significance (p = 0.08). Reaction time was significantly different between groups (p = 0.006) with CP slower. It was also significantly slower in the PA condition (p = 0.03) with groups combined.

Similar to the SRTT analysis, we correlated correct responses with reaction time for the WPT PA (unrewarded) condition and found a significant speed-accuracy trade-off for those data as well (r = -0.41; p = 0.045) indicating that higher accuracy was associated with lower response speed.

**Table 2. Mean changes in outcomes after training.**

| Condition | All | HV | CP | p-value |
|---|---|---|---|---|
| **SRTT Reaction Time (ms):** | | | | |
| Unrewarded Probable | 174.2 ± 292.7 | 122.4 ± 256.1 | 222.1 ± 325.7 | 0.40 |
| Unrewarded Improbable | 83.7 ± 403.1 | 91.0 ± 278.5 | 77.1 ± 503.8 | 0.93 |
| Rewarded Probable | 252.5 ± 362.6 | 168.7 ± 242.0 | 329.7 ± 447.4 | 0.27 |
| Rewarded, Improbable | 280.6 ± 589.9 | 273.5 ± 727.9 | 287.1 ± 458.3 | 0.96 |
| **SRTT Error Rate (%):** | | | | |
| Unrewarded Probable | 8.2 ± 15.6 | 2.8 ± 5.3 | 13.2 ± 20.1 | **0.09** |
| Unrewarded Improbable | 2.4 ± 2.9 | 1.4 ± 1.3 | 3.2 ± 3.6 | 0.11 |
| Rewarded Probable | 9.0 ± 15.5 | 2.8 ± 16.8 | 14.6 ± 19.1 | **0.054** |
| Rewarded Improbable | 2.8 ± 3.0 | 2.5 ± 2.9 | 3.2 ± 4.2 | 0.65 |
| **WPT Proportion Correct** | | | | |
| Paired Association (PA) | 73.5 ± 13.9 | 75.0 ± 14.2 | 72.2 ± 14.0 | 0.62 |
| Feedback (FB) | 67.3 ± 17.4 | 71.6 ± 13.6 | 63.4 ± 20.0 | 0.25 |
| **WPT Reaction Time** | | | | |
| Paired Association (PA) | 2807.7 ± 554.4 | 2504.6 ± 505.7 | 3087.4 ± 451.3 | **0.006***  |
| Feedback (FB) | 2477.9 ± 818.8 | 2350.9 ± 756.6 | 2595.0 ± 886.2 | 0.47 |

Results are shown for the entire group and the group with healthy volunteers (HV) or participants with cerebral palsy (CP) with associated p value of the independent t-test between groups. All p values< 0.10 are in **bold** and if significant (p <0.05), also indicated by *. SRTT = Serial reaction time task; WPT = Weather Prediction Test; FB = feedback; PA = paired association. The only significant difference between groups with CP or HV was for WPT reaction time which was slower in CP with notably no significant group difference in WPT proportion correct.

## Influence of gene score on SRTT and WPT outcomes

The primary aim of this study was to evaluate the influence of genetic variation on learning abilities. We first correlated the individual combined gene scores with values on each of the outcome measures in the group as a whole and then for each of the two subgroups. A significant correlation in WPT PA was found only in controls (r = -0.68; p = 0.02) such that those with higher gene scores had faster reaction times with the same trend seen between WPT FB reaction time and gene score (r = -0.53; p = 0.08). WPT FB correct scores with gene scores in controls also nearly reached significance (r = 0.57; p = 0.051). No correlations in CP alone were significant or showed strong trends, nor in the group as a whole. We also correlated all outcomes with the BDNF score and found no significant results for the group as a whole, or within groups, for SRTT reaction time and error rate, unrewarded and rewarded (0.80, 0.87, 0.65, 0.29, respectively) or for WPT FB and PA correct scores and reaction times (p = 0.18, 0.27, 0.29, 0.69, respectively).

The distribution of individual gene scores (GS) across participants were as follows: GS0 = 0; GS1 = 0CP, 2HV; GS2 = 5CP, 4HV; GS3 = 4CP, 3HV; GS4 = 3CP, 2HV; GS5 = 1 CP, 1HV). Given our small sample with few participants in the 0, 1, and 5 gene score categories, we subdivided participants into low (0, 1, 2; n = 11) and high (3, 4, 5; n = 14) gene score groups to evaluate whether these demonstrated any significant differences in any SRTT or WPT outcomes using independent t-tests as shown in Table 3.

The high gene group tended to have a lower error rate for the SRTT unrewarded probable condition (p = 0.08) and faster reaction time on WPT PA (p = 0.07) compared to the low gene group with the same pattern seen in all unrewarded outcomes. There was no significant difference between conditions (p = 0.72) and no interaction between gene group and condition (p = 0.60).

**Table 3. Means and standard deviations of change scores for all outcomes from probable conditions by low (0, 1, 2) and high (3, 4. 5) gene scores.**

| Outcome measure | Low gene score | High gene score | p-value |
|---|---|---|---|
| SRTT Reaction Time UP | 240.7 ± 352.7 | 122.0 ± 236.3 | 0.33 |
| SRTT Reaction Time RP | 273.3 ± 251.0 | 236.1 ± 444.7 | 0.81 |
| SRTT Error rate UP | 14.4 ± 18.7 | 3.4 ± 11.1 | **0.08** |
| SRTT Error Rate RP | 5.6 ± 15.9 | 11.6 ± 15.1 | 0.54 |
| WPT Correct Proportion PA | 63.3 ± 15.1 | 76.9 ± 12.3 | 0.18 |
| WPT Correct Proportion FB | 61.0 ± 17.5 | 72.3 ± 16.3 | 0.11 |
| WPT Reaction Time PA | 3031.9 ± 548.2 | 2631.5 ± 510.2 | **0.07** |
| WPT Reaction Time FB | 2542.5 ± 963.1 | 2427.1 ± 719.7 | 0.73 |

No between group differences were significant based on independent t-tests, but some trends favored the high gene group with the same pattern seen in all unrewarded outcomes. SRTT = Serial reaction time task; WPT = Weather Prediction Test; UP = unrewarded probable; RP = rewarded probable; FB = feedback; PA = paired association.

**Gene scores across unrewarded and rewarded probable conditions.** The true influence of gene scores only became apparent when we evaluated the unrewarded and rewarded conditions together. We had hypothesized that those who had a more 'optimal' level of DA brain transmission as reflected by higher gene scores would have better performance with training and would be less affected by rewards, while those with lower gene scores would have poorer baseline performance and show a greater response to rewards. The main finding for the GLM with repeated measures comparing rewarded and unrewarded conditions (within subjects) and high and low gene groups (between subjects) was a significant interaction between condition and gene group for SRTT error rate (p = 0.01). with no main effect found for either condition (p = 0.92) or gene group (p = 0.62) as shown in Table 4.

Fig 3 illustrates this major finding by presenting individual error rates for the two conditions for each gene group. In the unrewarded condition (baseline) the worst error rate scores were in participants with low gene scores with the best scores largely in those with high gene scores. Interestingly, the pattern reversed with rewards with those with low gene scores having a generally positive effect whereas those with high gene scores having a lesser and mean negative effect, *indicating that rewards had a differential effect depending on gene score*. SRTT Reaction time showed no significant main effect for condition (p = 0.41) or gene group (p = 0.46) and no interaction (p = 0.65).

WPT proportion correct also showed a significant difference between conditions, with the high gene score group having higher scores on both conditions. A significant interaction between condition and gene group was found but rewards in this case had a worse effect on the low gene group. The other significant finding was for WPT reaction time was a main effect

**Table 4. General linear model comparing unrewarded and rewarded conditions across high and low gene groups.**

| Outcome | Condition | Condition X Group | Group |
|---|---|---|---|
| SRTT RT UP vs. RP | 0.41 | 0.65 | 0.46 |
| SRTT Error Rate UP vs. RP | 0.92 | **0.01**\* | 0.65 |
| WPT PA vs. FB Correct | **0.07** | 0.60 | **0.08** |
| WPT PA vs. FB RT | **0.03**\* | 0.34 | 0.30 |

All trends (p < 0.10) are shown in **bold** and significant differences (p < 0.05) are indicated by

\*. SRTT = Serial reaction time task; WPT = Weather Prediction Test; UP = unrewarded probable; RP = rewarded probable; FB = feedback; PA = paired association; RT = reaction time.

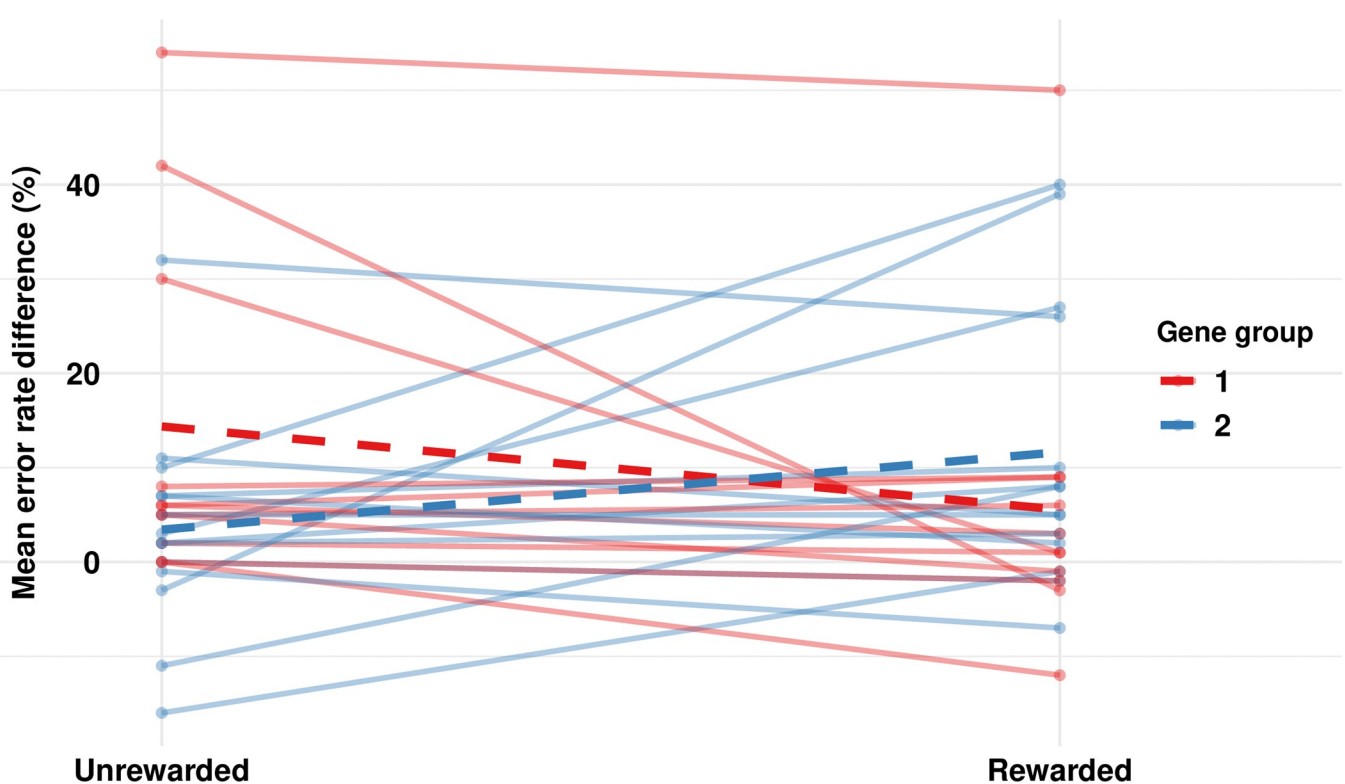

**Fig 3. Error rate change for unrewarded and rewarded training trials with darker lines showing the tendency for the low gene group to have poorer and the high gene group better error rate scores.** These show a differential response to rewards such that the low gene group tends to improve, and the high gene group tends to worsen as indicated by a significant interaction effect.

for condition (p = 0.03) with reaction time faster in the FB (rewarded) condition. There was no main effect for gene group (p = 0.30) and no interaction between gene group and condition (p = 0.34).

For the WPT, an Awareness score was calculated to test post hoc knowledge of the card associations with lower scores indicating greater awareness. The mean Awareness score in the control group was 61.0 ±7.26 and in the CP group was 94.2 ± 13.3 which was significantly worse (p = 0.04). An inverse correlation was observed between the FB proportion correct and Awareness scores (r = -0.53, p = 0.007; r = -0.44, p = 0.027, respectively) with greater awareness related to better performance.

### Effects of condition order, age and sex as a biological variable

Participants in each group alternately were assigned to perform the rewarded or unrewarded condition first. There was no consistent pattern in mean values depending on order and none of the differences in SRTT and WPT rewarded and unrewarded reaction time, SRTT error rate or WPT proportion correct with respect to order reached significance.

Age of participants was correlated with performance on all baseline (unrewarded) measures for the SRTT and the WPT. No significant correlations were seen between age and WPT outcomes; however, age was inversely correlated with the SRTT reaction time and error rate such that those who were older reacted more quickly and made fewer errors (SRTT reaction time: *r* = -0.47, p = 0.017; SRTT error rate: *r* = -0.52, p = 0.007). It is important to note that there was no age difference between the HV and CP groups (p = 0.80) or between low and high gene

score groups (p = 0.54), so any significant differences across groups could not be attributed to age differences.

Given evidence of sex differences in innate DA receptor expression and function across sexes (Williams et al. 2021), we evaluated results based on sex. The GLM comparing unrewarded and rewarded trials was repeated using sex as a between-group factor. No significant main effect for sex (n = 11 M; n = 14 F) was observed (p = 0.41) for SRTT outcomes, WPT FB (p = 0.59), or PA scores (p = 0.26).

## Relationships between the two tasks

As a final step to evaluate whether there were any relationships between the SRTT and the WPT tasks, we correlated the rewarded and unrewarded probable sequence error rates and reaction times to the WPT FB and PA proportion correct and reaction time scores. WPT proportion correct scores were not related to SRTT error rates; however, SRTT rewarded reaction time was directly associated with both the FB (r = 0.57, p = 0.003) and PA (r = 0.43, p = 0.03) WPT reaction times indicating that motor speed was related within individuals across these two tasks, but accuracy scores were not related.

## Discussion

Motivation for this study was based on the heterogeneity across rehabilitation outcomes commonly reported in children with CP, even those with similar types of brain injuries and functional levels, that remains poorly understood [7]. Neurorehabilitation for adults post-stroke or children with CP often requires the learning and practice of new motor skills or more coordinated movement strategies, i.e., motor learning. DA has been shown to have a strong relationship with early learning vs. retention of skilled motor performance in rodent models [43] but this relationship has been inadequately studied in humans. Studies in adults post-stroke, and in animal stroke models, have indicated that DA pathways are disrupted after the injury, and pharmacologic upregulation of DA receptors has been shown to improve motor recovery [12, 44, 45], thus suggesting that using medication or strategies that increase brain DA transmission may be an effective therapy.

The goal here was to explore the role of DA on learning in a cohort of children and young adults with and without CP using well-standardized tasks known to be related to DA and that contained both cognitive and motor aspects. In addition to known genetic influences, it has been hypothesized that DA transmission may be decreased in CP, as with stroke, as a consequence of the brain injury. A handful of studies have been conducted in CP which used medications to alter DA transmission [22, 23, 46] with some positive but generally insufficient results. It has been observed that levodopa may produce slower incremental positive changes over time in CP [47] rather than a dramatic nearly immediate response as in Parkinson's disease, perhaps because it improves the ability to learn motor skills which need to be acquired through repetitive practice typically over weeks and months. It is possible that the use of medication to increase DA neurotransmission for those with lower gene scores may be particularly useful when they are participating in intensive skill training programs since this may enhance skill acquisition.

The Pearson-Fuhrhop et al. [15] study was among the first to show the interaction between genetic variability and response to DA medication (L-dopa) in healthy adults that predictably affected performance on a motor learning task. This study influenced both the retrospective study by Diaz Heijtz et al. [24] as well as this investigation here, both involving cohorts with CP. Diaz Heijtz et al. [24] invited children with unilateral CP who had participated in constraint induced movement therapy trials months to years earlier to return for the collection of

saliva samples for genetic analysis of their DA gene and BDNF variants. In their study, they found a significant association between gene score and improvement in the Assisting Hand Assessment (AHA) as a result of the intensive training program. However, they acknowledged that gene score alone did not explain all between-subject variability in rehabilitation outcomes. Similar to our study, Diaz Heijtz et al. [24] found no association of BDNF with the AHA.

Worley et al. [48] also performed a similar genetic analysis to Pearson-Fuhrhop et al. [15] by analyzing six single nucleotide polymorphisms (SNPs) from five genes that influence DA neurotransmission directly and one SNP from the BDNF gene, associated with both diminished DA neurotransmission and reduced neuroplasticity, in a large cohort of extremely low birthweight infants some of whom were later diagnosed with CP. They found a significant relationship of gene score to the psychomotor developmental index (PDI) of the Bayley Scales of Infant Development, such that each 1 point increment on the combined gene score was associated with 1.37 points on the PDI after controlling for confounding clinical covariates such as cognitive development or ethnic differences in SNP proportions. Motor skill acquisition during development is a motor learning process, so this association makes logical sense. They further noted that this definitive but relatively small influence on motor development was consistent with most genetic associations with multifactorial complex phenotypes. In contrast to motor development, cognitive development was not related to gene scores. Gene scores were also not shown to be related to a predisposition towards a diagnosis of CP.

This is, to our knowledge, the first *prospective* study exploring genetic variations in DA and other plasticity-related genes and how they may relate to motor and/or cognitive learning abilities during training in individuals with and without CP. In addition to children and young adults with typical development, this study included those with both unilateral and bilateral CP with a range of etiologies and functional levels from I-III out of V on the mobility and manual function classifications [41, 49]. The SRTT involved a motor response of rapidly pressing the indicated computer key from among four possible options as well as an implicit cognitive task of learning a probable sequence. WPT was mainly a cognitive task that involved probabilistic learning although the response also involved a simple button press to indicate which of the two dichotomous outcomes (rain/sun; hot/cold) was predicted.

For the SRTT, we were able to demonstrate in the unrewarded condition that reaction time was faster for the probable sequence suggesting that learning of the probable sequence had occurred. For the WPT, we also demonstrated that participants successfully improved their outcome prediction with mean scores greater than chance (60+%). These results demonstrated that as a group, participants in either/both groups performed these tasks as designed with the exception of one participant in each group who found the tasks too difficult to understand or perform and had to withdraw. Some between group differences in performance were noted with those with CP having slower reaction times and higher error rates overall; however, the only significant difference when comparing the amount of change in response to training was on the WPT PA reaction time improvement which was significantly greater in CP. The minimal between group differences in the amount of change in training outcomes enabled us to justify the evaluation of gene scores relationships in the group as a whole, as well as separately.

When we addressed the primary aim which was to explore associations between individual variations in brain DA and/or plasticity-related genes and the ability to learn each of these two novel tasks in individuals with and without CP, we found that in the group as a whole and the group with CP, there were no significant correlations between the combined gene scores and outcomes. However, for the control group only, there was a significant correlation between WPT PA reaction time and gene score such that those with higher gene scores had faster reaction times with the same trend seen between WPT FB reaction time and gene score and WPT FB correct scores with gene scores in controls that were not seen in CP, or in the combined

sample, suggesting that the genetic relationships may have been somewhat obscured in CP due to disruptions in DA transmission related to the brain injury [23].

When comparing all unrewarded outcomes, which represented baseline learning abilities, across high vs. low gene score groups, it was noted that mean changes in all outcome measures for the low gene score group showed worse performance compared to the high gene score group, although differences did not reach significance.

The second aim of the study was to examine the effect of rewards on performance and how it might vary depending on gene score. Since it is well known that rewards increase DA brain transmission, these should provide greater or even exclusive benefit for those with lower baseline levels of DA. With unrewarded and rewarded conditions as the within subject factor and high and low gene scores groups as the between subject factor, a significant interaction was found for SRTT error rate by gene group as well as strong trends whereby scores for the low gene group tended to be lower in the unrewarded condition and increase with rewards, with the high gene score group showing better scores in the unrewarded condition that generally worsened with rewards. The greater training response in the unrewarded condition for those with higher gene scores mirrored those in the Diaz Heijtz et al. [24]. Further, the differential response to rewards with respect to gene scores mirrored the response to levodopa across gene score groups in the Pearson-Fuhrhop et al [15] study. Although our study only involved single session training for both the unrewarded and rewarded conditions, neurorehabilitation training often requires multiple sessions as was the case in the other two studies cited as comparators. However, the similarity of our findings with theirs was encouraging and added further support for the effects of genetic variation on learning given the differential responses here depending on gene group.

Although not a genetic study, Wilkinson et al. [40] temporarily disrupted learning on the SRTT in a cohort of healthy adults using repetitive theta burst TMS and found that rewards helped to "recover" their leaning abilities, similar to results here for those with poorer learning at baseline. We had not anticipated that the increase in DA as a result of rewards would be as great as the effect of administering levodopa which had a negative effect for some with high gene scores by increasing DA too much as indicated by the inverted U-shaped optimality curve reported for DA effectiveness [50, 51]. However, the significant interaction found here indicated that while too little DA may impair learning, too much DA in the form of rewards could also be detrimental to learning for some individuals.

A recent review by Zhao et al. [39] attempted to explain the observed inconsistencies in the effects of rewards on learning, with some studies showing positive effects in healthy adults as well as those post-stroke and others demonstrating inconsistent or even opposite results [52–54]. They conducted a review of the literature and evaluated responses to rewards based on the type of motor learning task and the different stages of learning in hopes of identifying a consistent pattern of when rewards were or were not helpful. The authors stated that two main types of motor learning have been the focus of research: motor adaptation and motor sequence learning, and for the latter, the most commonly used paradigm is the SRTT involving arm reaching or finger pressing and characterized by shifts in the speed-accuracy relationship. In their review of studies, they made several observations relevant to our results. The cited a study by Anderson et al. [55] that demonstrated greater learning in a motor sequence task when monetary incentives were used as rewards by enhancing motivation. A different study by Sporn et al. [56] agreed that monetary awards alone appear to speed up movement time, whereas they claimed that the addition of corrective feedback primarily improved performance. In our study, SRTT reaction time increased markedly with rewards with no significant increase in errors in the group as a whole when monetary rewards and corrective feedback were provided simultaneously. On the WPT, when monetary rewards were provided during

training along with corrective feedback, versus passively demonstrating the correct answers after each card combination, both speed and performance changes were worse.

Zhao et al. were unsuccessful in explaining the effect of reward due to task type and instead concluded that even though money is a strong reward, high individual variability in responsiveness prevailed and largely explained the disparity in data across studies. They stated that identifying the still unknown contributing factors to this variability is imperative for determining whether to use rewards on an individual basis, i.e., towards personalized rehabilitation. Interestingly, they did not suggest that the individual variability may be genetic in origin as demonstrated in our study.

We had initially anticipated that the main result of the SRTT would have been increased reaction time as the sequence was implicitly "learned" over time and would have been the outcome most related to genetic differences in DA levels. Instead, error rates more clearly demonstrated differences in performance based on gene scores and differential responses with rewards across gene score groups even though these tended to *increase* with training in the baseline unrewarded probable condition. Interestingly, a recent study by Leow et al. [57] examined the effect of DA administration on a target aiming task in healthy adults and found that increasing DA had a primary effect of improving accuracy with the trade-off of lengthening reaction times. Our results were in contrast to theirs with reaction time increasing with training in both tasks, and in fact participant reaction time scores were correlated across tasks. The significantly higher SRTT unrewarded error rate in CP controls was correlated inversely with reaction time (i.e., demonstrated a speed-accuracy trade-off). The explanation for why these participants seemed to prioritize speed over accuracy in the SRTT is not entirely clear because this was the case for both the unrewarded and rewarded conditions. It is also surprising because rewards for the SRTT were based on accuracy and were only given if they had given the correct response within the allotted time, even though they were instructed to respond "as quickly and accurately as possible". It is important to note that even though error rates generally increased with training, the relative differences in error rates varied predictably in relation to high or low gene scores and in their respective responses to rewards.

Similar to others, we evaluated the effects of specific genes which although preliminary, indicated a stronger role of DRD 1 for the WPT results. Nikolova et al. [58] investigated the additive effects of five polymorphisms affecting DA signaling on reward-related ventral striatum (VS) reactivity, measured with BOLD fMRI, in 69 Caucasian adults. Their task was designed to specifically engage the ventral striatum and involved a card guessing game where they were shown a card and had to respond within 3 seconds whether it was higher or lower than 5, after which they received positive or negative feedback with associated monetary rewards depending on whether they were correct or not. The results indicated that this combined gene score explained 10.9% of the individual variability in reactivity, which the authors suggested would likely have been obscured when trying to identify even smaller effects in individual SNPs. Their work further elucidated the neural and genetic mechanisms underlying DA brain transmission in response to rewards. It further supported the small but significant statistical results of the predictive relationships of gene score with motor learning abilities shown here as well as by Diaz Heijtz et al. [24], Worley et.al. [48], and the response to DA manipulation by medication as in Pearson-Fuhrhop et al. [15] study or rewards as shown in this study.

Worley et al. [48] presented a thoughtful discussion of the many advantages of using polygenic risk scores, such as the combination of DA genes now utilized across studies, instead of sequential testing of associations of multiple individual SNPs. They proposed that these analyses are more consistent with the genetic underpinnings of complex phenotypes, which are thought to be polygenic, i.e., influenced by many genes by combining multiple variants, each of which may have a small effect and a skewed distribution, into one predictor variable that

measures a larger total effect thereby increasing statistical power. They further stated that this approach reduces the number of hypotheses tested thus greatly reducing the size of the sample needed when testing individual variants; and finally, the predictive model can be updated by adjusting the weighting of different variants or adding new genetic variants that relate to the phenotype of interest as they are discovered.

The WPT is a different type of task than the SRTT that involves probabilistic learning with associated differences in DA-related effects. There is evidence that those with a brain injury or disorder with depletion in some DA circuits (e.g. Parkinson's disease and ADHD) may have impairments in responses to feedback [59, 60]; whereas, they can learn the task similarly to non-injured controls in the PA condition. Consistent with that, our results showed similar results for CP and controls for the PA condition (p = 0.58). In the FB condition, both groups were slightly worse, but only the difference in CP neared significance (p = 0.051), which may have been affected by DA depletion from the brain injury. Reaction time was significantly faster for the FB compared to the PA condition, but neither participant nor gene group accounted for this difference; however, those with higher gene scores had a similar pattern to SRTT outcomes of better although not significant mean performance and reaction time scores in both WPT conditions. One explanation for the WPT results for genetic variability not being apparent as for the SRTT may be related to the findings in the Worley et al. study which only found an influence of genetic variability in motor development and not cognitive development, similar to results in animal studies. They concluded that the neural circuitry involved in cognition and cognitive development may not be as closely linked to DA neurotransmission. The WPT was utilized here as a measure of cognitive ability as compared to the SRTT which in contrast has a motor sequence learning component. Another potential explanation for the difference in results across tasks is the conclusion from the Alm review [61] that one of the core functions of central DA involves the learning and execution of automatized sequences of movements which is a key component of the SRTT but not the WPT.

## Limitations

The major limitation of this study was the small sample size that limited the power to find significant group and condition differences and correlations. In particular, the limited numbers in several of the gene score groupings made it necessary to subdivide participants into two gene groups instead of six possible gene scores (0–5). Still there were several important significant findings, particularly the SRTT error rate interaction of condition (rewarded and unrewarded) by low vs. high gene group which supported the initial hypotheses that those with lower gene scores would tend to have poorer learning at baseline and respond more positively to rewards.

## Conclusion

This study demonstrated that individual genetic variability affected learning outcomes which has important relevance for rehabilitation as shown by the predictability of a combined gene score on learning abilities as well as on response to rewards, or potentially on other strategies such as DA-modifying medications. For children with CP, genetic influences start in early motor development and should be considered as critical new information for developing more personalized approaches in clinical neurorehabilitation. Specifically, if those at greater genetic risk for slower development or slower learning rates could be identified sooner, their motor outcomes could potentially be enhanced through training approaches or medications that upregulate DA, especially during periods of rapid development or more intensive motor training.

## Supporting information

**S1 Table. Calculation of individual gene scores.**
(DOCX)

**S2 Table. Effect of individual genes.**
(DOCX)

**S1 Fig.** a. **Mean change in reaction time per block during training for the group with CP and controls (HV) for the two probable sequences in the unrewarded and rewarded conditions** Baseline (unrewarded) scores are relatively worse but show greater changes over time in CP, and there appears to be a baseline shift towards faster reaction times in the rewarded condition. b. **Mean change in error raate per block during training for the group with CP and controls (HV) for the two probable sequences for the unrewarded and rewarded conditions** Relatively worse baseline scores and increasing errors observed in CP that are lower in the Rewarded condition. The HV group shows a generally flatter pattern and one that shows a slightly negative rather than positive effect from rewards.
(ZIP)

## Acknowledgments

We acknowledge the help of Dr. Leonora Wilkinson who during her post-doctoral fellowship at NINDS/NIH developed the computer versions of the tasks utilized here.

## Author Contributions

**Conceptualization:** Hans Forssberg, Diane L. Damiano.

**Data curation:** Barrett Dryden, Jesse Matsubara, Diane L. Damiano.

**Formal analysis:** Barrett Dryden, Jesse Matsubara, Diane L. Damiano.

**Investigation:** Jesse Matsubara, Diane L. Damiano.

**Methodology:** Eric Wassermann, Diane L. Damiano.

**Project administration:** Jesse Matsubara, Diane L. Damiano.

**Resources:** Eric Wassermann, Diane L. Damiano.

**Software:** Barrett Dryden, Jesse Matsubara.

**Supervision:** Diane L. Damiano.

**Visualization:** Jesse Matsubara.

**Writing – original draft:** Barrett Dryden, Diane L. Damiano.

**Writing – review & editing:** Eric Wassermann, Hans Forssberg, Diane L. Damiano.

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
