## [Decision Letter · Decision Letter 0]

11 Sep 2024

PONE-D-24-30058Effect of individual variations in genes related to dopamine brain transmission on performance with and without rewards during motor sequence and probabilistic learning tasks in children and young adults with and without Cerebral PalsyPLOS ONE

Dear Dr. Damiano,

Thank you for submitting your manuscript to PLOS ONE. After careful consideration, we feel that it has merit but does not fully meet PLOS ONE’s publication criteria as it currently stands. Therefore, we invite you to submit a revised version of the manuscript that addresses the points raised during the review process.

We look forward to receiving your revised manuscript.

Kind regards,

Alexey Kuznetsov

Academic Editor

PLOS ONE

Journal Requirements:

“This work was funded by the Intramural Research Program at the National Institutes of Health Clinical Center (Protocol # 16-CC-0149). We also acknowledge the help of Dr. Leonora Wilkinson who developed the computer versions of the tasks utilized here.”

3. In the online submission form you indicate that your data is not available for proprietary reasons and have provided a contact point for accessing this data. Please note that your current contact point is a co-author on this manuscript. According to our Data Policy, the contact point must not be an author on the manuscript and must be an institutional contact, ideally not an individual. Please revise your data statement to a non-author institutional point of contact, such as a data access or ethics committee, and send this to us via return email. Please also include contact information for the third party organization, and please include the full citation of where the data can be found.

4. We notice that your supplementary tables are included in the manuscript file. Please remove them and upload them with the file type 'Supporting Information'. Please ensure that each Supporting Information file has a legend listed in the manuscript after the references list.

**Additional Editor Comments:**

Specifically, corrections required and answers to the questions raised by the reviewers are recommended to be incorporated into the manuscript before publishing.

Reviewers' comments:

Reviewer's Responses to Questions

**Comments to the Author**

1. Is the manuscript technically sound, and do the data support the conclusions?

Reviewer #1: Yes

Reviewer #2: Yes

2. Has the statistical analysis been performed appropriately and rigorously? 

Reviewer #1: Yes

Reviewer #2: Yes

3. Have the authors made all data underlying the findings in their manuscript fully available?

Reviewer #1: Yes

Reviewer #2: Yes

4. Is the manuscript presented in an intelligible fashion and written in standard English?

Reviewer #1: Yes

Reviewer #2: Yes

5. Review Comments to the Author

Reviewer #1: The manuscript examines the influence of dopamine-related genetic variability on learning in young individuals with and without cerebral palsy (CP). The study calculated individual dopamine-related gene scores and compared these to performance on two tasks: an implicit sequence learning task and a probabilistic classification task. Both tasks were administered in unrewarded and rewarded conditions, the latter known to increase dopamine levels. The key finding was an interaction between gene score and condition for the sequence learning task: individuals with low gene scores performed poorly without rewards but improved with rewards, while those with high gene scores showed the opposite pattern.

The study is exceptionally well-written and demonstrates a comprehensive understanding of the literature. Despite the small sample size, it represents a significant milestone by highlighting the potential for personalized neuro-rehabilitation approaches, showing how dopamine-related genetic differences can influence motor skill learning in young individuals with CP.

I just have a few minor comments:

1. On Page 6, please ensure the correct usage of 'Brain-derived neurotrophic factor (BDNF).'

2. On Page 20, correct the name 'Diaz Heist' to 'Diaz Heijtz.'

3. Please add a footnote to Table I with brief explanations for GMFCS and MACS to help general readers understand these classifications.

Reviewer #2: The authors studied the relationship between dopamine genes, expressed collectively as a gene score, and motor learning using a 2x2 design, consisting of an implicit sequence learning task and a probabilistic classification task, each administered in an unrewarded condition and a rewarded one. Study subjects had CP (n=13) or were healthy controls (n=12). The main finding was an interaction between gene score and condition for the sequence task, such that those with low gene scores were poorer learners without rewards but responded positively to rewards; whereas the converse was true for those with high gene scores. The authors conclude that that genetic variability related to dopamine may influence neurorehabilitation outcomes and could potentially be modulated using rewards or medications for those with poorer learning at baseline, which can promote a more personalized approach to patient care.

The report is well written, and the conclusions are justified by study results.

Were the main significant results significant in CP, healthy controls, or both?

The verb tense jumps around between present and future, which could confuse some readers.

Subjects were age 5-25, thus: did results vary by subject age?

Was there an order effect with respect to order of reward condition and order of task?

Line 305 is “CP vs. TD group”, and the term “TD” is used later. But “TD” is not defined, but it should be.

Line 367 Notes that “The true influence of gene scores only became apparent when we evaluated the unrewarded and rewarded conditions together.” This indicates that the paragraph with lines 340-351 evaluated the unrewarded and rewarded conditions separately, but this is not clear when reading that paragraph. The authors are asked to clarify these points to ease reader comprehension.

One limitation that the authors are encouraged to discuss is that the current findings are based on single session learning paradigms. However, neurorehabilitation typically occurs over many sessions, and so some uncertainty exists as to how current findings extrapolate.

6. PLOS authors have the option to publish the peer review history of their article (what does this mean?). If published, this will include your full peer review and any attached files.

Reviewer #1: No

Reviewer #2: No

---

## [Author Response · Author response to Decision Letter 0]

24 Sep 2024

PONE-D-24-30058

Effect of individual variations in genes related to dopamine brain transmission on performance with and without rewards during motor sequence and probabilistic learning tasks in children and young adults with and without Cerebral Palsy

Thanks to the Editor and to the reviewers for describing what changes were required or recommended and for the overall positive response to the manuscript. We have addressed each comment as detailed below with our responses in italics.

Dear Dr. Damiano,

Thank you for submitting your manuscript to PLOS ONE. After careful consideration, we feel that it has merit but does not fully meet PLOS ONE’s publication criteria as it currently stands. Therefore, we invite you to submit a revised version of the manuscript that addresses the points raised during the review process.

Journal Requirements: 

We have carefully reviewed all instructions and have made modifications throughout the manuscript to comply with all.

“This work was funded by the Intramural Research Program at the National Institutes of Health Clinical Center (Protocol # 16-CC-0149). We also acknowledge the help of Dr. Leonora Wilkinson who developed the computer versions of the tasks utilized here.”

Please remove any funding-related text from the manuscript and let us know how you would like to update your Funding Statement. 

The acknowledgement was updated in the text. The funding statement was updated in the Cover letter.

3. In the online submission form you indicate that your data is not available for proprietary reasons and have provided a contact point for accessing this data. Please note that your current contact point is a co-author on this manuscript. According to our Data Policy, the contact point must not be an author on the manuscript and must be an institutional contact, ideally not an individual. Please revise your data statement to a non-author institutional point of contact, such as a data access or ethics committee, and send this to us via return email. Please also include contact information for the third party organization, and please include the full citation of where the data can be found.

We have submitted this to the Dryad data repository on 9/18/24 with the following confirmation:

Relating genetic variations in dopamine brain transmission to task performance with and without rewards submitted with DOI https://doi.org/10.5061/dryad.qnk98sfs5. There may be a delay for processing before the item is available.

4. We notice that your supplementary tables are included in the manuscript file. Please remove them and upload them with the file type 'Supporting Information'. Please ensure that each Supporting Information file has a legend listed in the manuscript after the references list.

We have created Supporting information files and listed the legends in the manuscript after the references.

All are correct – two needed to be added in response to a reviewer’s comment.

5. Review Comments to the Author

Reviewer #1: The manuscript examines the influence of dopamine-related genetic variability on learning in young individuals with and without cerebral palsy (CP). The study calculated individual dopamine-related gene scores and compared these to performance on two tasks: an implicit sequence learning task and a probabilistic classification task. Both tasks were administered in unrewarded and rewarded conditions, the latter known to increase dopamine levels. The key finding was an interaction between gene score and condition for the sequence learning task: individuals with low gene scores performed poorly without rewards but improved with rewards, while those with high gene scores showed the opposite pattern.

The study is exceptionally well-written and demonstrates a comprehensive understanding of the literature. Despite the small sample size, it represents a significant milestone by highlighting the potential for personalized neuro-rehabilitation approaches, showing how dopamine-related genetic differences can influence motor skill learning in young individuals with CP.

I just have a few minor comments:

1. On Page 6, please ensure the correct usage of 'Brain-derived neurotrophic factor (BDNF)

Thank you for noticing this. We mixed up the variants for COMT and BDNF and have now corrected this.

2. On Page 20, correct the name 'Diaz Heist' to 'Diaz Heijtz.' 

Done

3. Please add a footnote to Table I with brief explanations for GMFCS and MACS to help general readers understand these classifications.

We have now added a brief description to the text with references for each and a footnote in the table to spell out these acronyms.

Reviewer #2: The authors studied the relationship between dopamine genes, expressed collectively as a gene score, and motor learning using a 2x2 design, consisting of an implicit sequence learning task and a probabilistic classification task, each administered in an unrewarded condition and a rewarded one. Study subjects had CP (n=13) or were healthy controls (n=12). The main finding was an interaction between gene score and condition for the sequence task, such that those with low gene scores were poorer learners without rewards but responded positively to rewards; whereas the converse was true for those with high gene scores. The authors conclude that that genetic variability related to dopamine may influence neurorehabilitation outcomes and could potentially be modulated using rewards or medications for those with poorer learning at baseline, which can promote a more personalized approach to patient care.

The report is well written, and the conclusions are justified by study results.

Were the main significant results significant in CP, healthy controls, or both?

The main result analyzing gene scores with performance in the rewarded and unrewarded conditions with both groups combined showed a significant interaction between gene score group and SRTT error rates.

Other significant results included:

• SRTT reaction time improvement was greater for the probable compared to the improbable sequence (as expected)

• SRTT error rate increased over time and more for the probable sequence for groups combined

• SRTT error rate higher in CP

• WPT reaction time slower in CP

• WPT reaction time slower in PA compared to FB for groups combined

It is not surprising that those with CP tended to move more slowly or make more key press errors, but they had similar learning in response to training as those with HV, and the effect of rewards was not seen in the group as a whole or in either subgroup until gene scores were introduced as a factor.

The verb tense jumps around between present and future, which could confuse some readers.

Thanks, we have now carefully checked through the whole document and made changes throughout the text to be more consistent. 

Subjects were age 5-25, thus: did results vary by subject age?

We had correlated the results with age and found no relationship with the WPT cognitive task but there was a moderate relationship between age and SRTT reaction time and error rate with younger being slower (r = 0,47; p = 0.02) and making more errors (r = 0.52; p = 0.01) which is not surprising because motor coordination and speed improve with age. We chose not include this for 2 reasons: 1. Long length of the manuscript already and 2. Age did not differ between groups so main results were not impacted.

If you think these should be included, we would be happy to do so.

Was there an order effect with respect to order of reward condition and order of task?

The order was randomized across subjects to control for this. And as an additional step, when we did a further check and compared results for those who did each first with those who did each second, there was no significant difference.

Line 305 is “CP vs. TD group”, and the term “TD” is used later. But “TD” is not defined, but it should be.

Thanks for noticing this. We should have used HV instead as had been used throughout the rest of the manuscript to indicate the control group. “TD” has now been removed and replaced by “HV”.

Line 367 Notes that “The true influence of gene scores only became apparent when we evaluated the unrewarded and rewarded conditions together.” This indicates that the paragraph with lines 340-351 evaluated the unrewarded and rewarded conditions separately, but this is not clear when reading that paragraph. The authors are asked to clarify these points to ease reader comprehension.

The paragraph 340-351 had not yet included gene scores as a factor, so the effect of rewards (which was subsequently found to be positive in those with low gene scores and the opposite in those with high gene scores) was washed out when analyzing the group as a whole or CP or HV alone, as was also seen in the Pearson-Fuhrhop results until they looked at the gene scores.

One limitation that the authors are encouraged to discuss is that the current findings are based on single session learning paradigms. However, neurorehabilitation typically occurs over many sessions, and so some uncertainty exists as to how current findings extrapolate

This is a good point. As suggested, we noted this difference in the discussion in lines 530-534 but added that our findings were encouragingly similar to those with longer training.

Done

---

## [Decision Letter · Decision Letter 1]

20 Oct 2024

PONE-D-24-30058R1Effect of individual variations in genes related to dopamine brain transmission on performance with and without rewards during motor sequence and probabilistic learning tasks in children and young adults with and without Cerebral PalsyPLOS ONE

Dear Dr. Damiano,

Thank you for submitting your manuscript to PLOS ONE. After careful consideration, we feel that it has merit but does not fully meet PLOS ONE’s publication criteria as it currently stands. Therefore, we invite you to submit a revised version of the manuscript that addresses the points raised during the review process. Specifically, two minor requests remain:

Please provide a very brief summary of results regarding group effect (CP vs. healthy controls) and age effect, in the Results section. Each can be quite concise. I understand space constraints, and that complex results can sometimes blur key findings, but future readers of the manuscript will likely also have questions about these two effects.

Regarding ordering effect, the authors note “when we did a further check and compared results for those who did each first with those who did each second, there was no significant difference.” Randomization doesn’t always solve things completely, as the authors know, especially with smaller sample sizes, and so reporting this finding will increase the impact of the overall findings. As such, can the authors please add this result to the manuscript? We encourage you to address the requests as they should further improve the quality of the manuscript.

We look forward to receiving your revised manuscript.

Kind regards,

Alexey Kuznetsov

Academic Editor

PLOS ONE

Journal Requirements:

Reviewers' comments:

Reviewer's Responses to Questions

**Comments to the Author**

1. If the authors have adequately addressed your comments raised in a previous round of review and you feel that this manuscript is now acceptable for publication, you may indicate that here to bypass the “Comments to the Author” section, enter your conflict of interest statement in the “Confidential to Editor” section, and submit your "Accept" recommendation.

Reviewer #2: All comments have been addressed

2. Is the manuscript technically sound, and do the data support the conclusions?

Reviewer #2: Yes

3. Has the statistical analysis been performed appropriately and rigorously? 

Reviewer #2: Yes

4. Have the authors made all data underlying the findings in their manuscript fully available?

Reviewer #2: Yes

5. Is the manuscript presented in an intelligible fashion and written in standard English?

Reviewer #2: Yes

6. Review Comments to the Author

Reviewer #2: The authors’ response are overall acceptable. Two minor requests remain:

Please provide a very brief summary of results regarding group effect (CP vs. healthy controls) and age effect, in the Results section. Each can be quite concise. I understand space constraints, and that complex results can sometimes blur key findings, but future readers of the manuscript will likely also have questions about these two effects.

Regarding ordering effect, the authors note “when we did a further check and compared results for those who did each first with those who did each second, there was no significant difference.” Randomization doesn’t always solve things completely, as the authors know, especially with smaller sample sizes, and so reporting this finding will increase the impact of the overall findings. As such, can the authors please add this result to the manuscript?

7. PLOS authors have the option to publish the peer review history of their article (what does this mean?). If published, this will include your full peer review and any attached files.

Reviewer #2: No

---

## [Author Response · Author response to Decision Letter 1]

31 Oct 2024

Response to Reviewers:

Thanks for your review. We have included the reviewer’s comments and our responses for each below.

Reviewer #2: Two minor requests remain:

Please provide a very brief summary of results regarding group effect (CP vs. healthy controls) and age effect, in the Results section. Each can be quite concise. I understand space constraints, and that complex results can sometimes blur key findings, but future readers of the manuscript will likely also have questions about these two effects.

We have now added a statement to the results summarizing the single difference found between groups. This had also been stated in the discussion in the previous version of the manuscript. We have also summarized the results of the analysis of age effects to the section where we had previously examined the effects of sex. 

Regarding ordering effect, the authors note “when we did a further check and compared results for those who did each first with those who did each second, there was no significant difference.” Randomization doesn’t always solve things completely, as the authors know, especially with smaller sample sizes, and so reporting this finding will increase the impact of the overall findings. As such, can the authors please add this result to the manuscript?

We have now included the data analyses for each outcome depending on order, as shown in the table included in the Response to reviewers document that was downloaded with the revised manuscript. We have summarized this in the results along with the results for age and sex, stating that there was no significant effect of order. We have also added a statement on the additional analyses done in the statistical analysis section.

---

## [Editor Report · Decision Letter 2]

6 Nov 2024

Effect of individual variations in genes related to dopamine brain transmission on performance with and without rewards during motor sequence and probabilistic learning tasks in children and young adults with and without Cerebral Palsy

PONE-D-24-30058R2

Dear Dr. Damiano,

We’re pleased to inform you that your manuscript has been judged scientifically suitable for publication and will be formally accepted for publication once it meets all outstanding technical requirements.

Kind regards,

Alexey Kuznetsov

Academic Editor

PLOS ONE

---

## [Editor Report · Acceptance letter]

8 Nov 2024

PONE-D-24-30058R2 

PLOS ONE

Dear Dr. Damiano, 

I'm pleased to inform you that your manuscript has been deemed suitable for publication in PLOS ONE. Congratulations! Your manuscript is now being handed over to our production team.

Kind regards, 

on behalf of

Dr. Alexey Kuznetsov 

Academic Editor

PLOS ONE